# TASK-AWARE PRIVACY PRESERVATION FOR MULTI-DIMENSIONAL DATA

## ABSTRACT

Local differential privacy (LDP), a state-of-the-art technique for privacy preservation, has been successfully deployed in a few real-world applications. In the future, LDP can be adopted to anonymize richer user data attributes that will be input to more sophisticated machine learning (ML) tasks. However, today's LDP approaches are largely *task-agnostic* and often lead to sub-optimal performance – they will simply inject noise to all data attributes according to a given privacy budget, regardless of what features are most relevant for an ultimate task. In this paper, we address how to significantly improve the ultimate task performance for multi-dimensional user data by considering a task-aware privacy preservation problem. The key idea is to use an encoder-decoder framework to learn (and anonymize) a *task-relevant* latent representation of user data, which gives an analytical near-optimal solution for a linear setting with mean-squared error (MSE) task loss. We also provide an approximate solution through a learning algorithm for general nonlinear cases. Extensive experiments demonstrate that our task-aware approach significantly improves ultimate task accuracy compared to a standard benchmark LDP approach while guaranteeing the same level of privacy.

## 1 INTRODUCTION

In recent years, there has been a tremendous growth in the volume of available data for machine learning (ML) tasks, leading to increasing emphasis on protecting user privacy. Differential privacy (DP) (Dwork et al., 2006; 2014) is a state-of-the-art technique for data privacy, and its local variant – local differential privacy (LDP) (Kasiviswanathan et al., 2011) – provides stronger privacy guarantees for individual users without dependence on any trusted third party. In practice, LDP has been successfully deployed in products of companies like Google (Erlingsson et al., 2014), Apple (Differential Privacy Team), and Microsoft (Ding et al., 2017) for some basic frequency or histogram estimation tasks where raw user data is restricted to an $n$-bit discrete variable.

In the future, LDP has the promising potential to be adopted in more complex scenarios (Hassan et al., 2019; Dankar & El Emam, 2013; Zhao et al., 2014; Cortés et al., 2016) (e.g., health care, power grids, Internet of Things) that feature richer user data attributes that feed into more sophisticated downstream ML tasks. In such cases, today's standard task-agnostic LDP approaches may not be ideal. For example, consider complex user data that must be anonymized before passing it into a ML task function, such as a neural network classifier for credit scores. A standard approach would be to simply perturb the data by adding artificial noise whose scale depends on the sensitivity of user data (i.e. worst-case variation among a user population) and a given privacy budget, *regardless* of what ultimate task the anonymized data will be used for. However, as the dimension and variability of user data inevitably grows, today's methods would generally have to increase the scale of noise to provide the same LDP guarantee, even though many data attributes might be highly variable across a user population, but minimally *relevant* for a task. As a consequence, one often adds excessive noise to all data attributes, which can severely degrade an ultimate task's performance.

To address these challenges, this paper introduces a fundamentally different *task-aware* LDP approach. Our method improves the performance of ML tasks that operate on multi-dimensional user data while still guaranteeing the same levels of privacy. Our key technical insight is to characterize the dependence of task performance on various user data attributes, which guides how we *learn* a concise, task-relevant encoding (i.e. latent representation) of user data. Then, for the same privacy

budget, we can directly expose and perturb only the task-relevant encoding rather than raw user data, which often allows us to add less noise and thereby improve task accuracy (see Section 2 for a concrete example). Crucially, user privacy is guaranteed under the same privacy budget according to the post-processing immunity of DP (Dwork et al., 2014) (i.e., one cannot make the output of a privacy algorithm less differentially private without additional knowledge). As such, an adversary cannot decode the anonymized latent representation to reduce the level of privacy. Our method allows us to learn and expose only high-valued data attributes and flexibly adjust their signal-to-noise ratio based on their importance to a task. Moreover, when different data attributes are inter-dependent, task-aware LDP preservation is even more promising in that we can consider the utilities of the underlying orthogonal bases through principal component analysis (PCA) (Dunteman, 1989), instead of the raw data attributes.

**Contributions.** In light of prior work, our contributions are three-fold. First, we propose a task-aware privacy preservation problem in which the effect of noise perturbation to preserve LDP is effectively considered, based on an encoder-decoder framework (Section 3). Second, in terms of task-aware privacy preservation, we obtain an analytical near-optimal solution for a linear setting and MSE task loss, and provide a heuristic learning algorithm for more general settings (Section 4). Third, we validate the effectiveness of our task-aware approach through three real-world experiments, which show our task-aware approach outperforms the benchmark approaches on overall task loss under various LDP budgets by as much as $73.0\%$ (Section 5). All the proofs are given in the Appendix.

**Related Work.** 1) Utility maximization in DP/LDP. Most theoretical DP/LDP research either provides a utility upper bound for a given privacy budget under some weak assumptions (Alvim et al., 2011; Kenthapadi et al., 2012; Duchi et al., 2013; Makhdoumi & Fawaz, 2013; Hardt & Talwar, 2010; Wang et al., 2019b; Acharya et al., 2020), or designs optimal privacy preservation mechanisms in some specific use cases (McSherry & Talwar, 2007; Wasserman & Zhou, 2010; Friedman & Schuster, 2010; Xiao et al., 2010; Thakurta & Smith, 2013; Kairouz et al., 2014; Geng & Viswanath, 2015; Liu et al., 2016; Yiwen et al., 2018; Joseph et al., 2019; Gondara & Wang, 2020; Wang et al., 2020). To the best of our knowledge, with respect to *multi-dimensional user data*, Murakami & Kawamoto (2019), Wang et al. (2019a) and Chen et al. (2021) are the only three similar works to ours. Murakami & Kawamoto (2019) develops a utility-maximizing LDP framework under the assumption that some data attributes may not be privacy-sensitive, and hence the utility improvement is mainly achieved by providing privacy guarantees for only a subset of attributes. Wang et al. (2019a) mainly focuses on developing novel LDP mechanisms for multi-dimensional data with an objective of minimizing the worst-case noise variance, which naturally improves utility. Chen et al. (2021) preserves DP for images by adding noise to a latent representation learned through back-propagation of task loss, where the effect of noise perturbation is not considered. The key differences of our work are: i) we don't make additional assumptions on the sensitivity of user data, ii) our task-aware approach achieves a better task performance than standard LDP benchmarks by directly studying the *dependencies* between the task objective and different attributes of user data, and iii) we effectively capture the effect of noise perturbation resulting from privacy requirements. 2) End-to-end (E2E) learning. There has been a wide variety of works training a cascade of deep neural networks (DNNs) through E2E learning, where a task-specific output is directly predicted from the raw inputs (Muller et al., 2006; Wang et al., 2012; Donti et al., 2017; Zhou & Tuzel, 2018; Amos et al., 2018). Our work also follows such a practice, but introduces an LDP guarantee while improving the task performance. 3) DP in deep learning. The popularity of deep learning also draws great attention to DP preservation therein (Song et al., 2013; Shokri & Shmatikov, 2015; Abadi et al., 2016; Phan et al., 2016; Papernot et al., 2016; Phan et al., 2017; McMahan et al., 2018; Wang et al., 2018; Phan et al., 2019; Arachchige et al., 2019; Liu et al., 2020; Mireshghallah et al., 2020; Bu et al., 2020). Our work is fundamentally different. Instead of preserving LDP *during the learning process*, we use learning as a tool to find the salient representation that improves the task performance under a given privacy budget. In other words, we don't perturb the gradient for back-propagation but perturb the representation to guarantee LDP. Furthermore, we don't specifically deal with privacy preservation during the *offline* training process, which requires some ground truth user data (e.g., from a small set of consenting volunteers). However, LDP of user data is guaranteed after a trained model is deployed *online*.

## 2 BACKGROUND AND MOTIVATING EXAMPLE

$\epsilon$**-LDP (Kasiviswanathan et al., 2011).** Let $x \in \mathbb{R}^n$ be an individual data sample, and $X$ be the domain of $x$, which is assumed to be a compact subset of $\mathbb{R}^n$. A randomized algorithm $\mathcal{M} : X \mapsto \mathbb{R}^d$ is said to satisfy $\epsilon$-LDP with *privacy-budget* $\epsilon > 0$, if

$$\Pr[\mathcal{M}(x) \in \mathcal{S}] \le e^\epsilon \Pr[\mathcal{M}(x') \in \mathcal{S}], \quad \forall x, x' \in X, \mathcal{S} \subseteq \text{im } \mathcal{M}. \tag{1}$$

Essentially, when $\epsilon$ is small, one cannot readily differentiate whether the input of $\mathcal{M}$ is an individual user $x$ or $x'$ based on $\mathcal{M}$'s outcome.

**Laplace mechanism (Dwork et al., 2014).** To release a sensitive function $g : X \mapsto \mathbb{R}^d$ under $\epsilon$-LDP, $\forall \epsilon > 0$, the Laplace mechanism is a widely used mechanism which adds Laplace noise to function $g$:

$$\mathcal{M}_{\text{Lap}}(x, g, \epsilon) = g(x) + \text{Lap}^d(\mu = 0, b = \frac{\Delta_1 g}{\epsilon}), \tag{2}$$

where $\text{Lap}^d(\mu, b)$ is a $d$-dimensional vector whose elements are i.i.d. Laplace random variables with mean $\mu$ and scale $b$, which leads its variance to be $2b^2$. For concreteness, the analysis and evaluation of this paper focus on the Laplace mechanism although the central idea of learning task-relevant data representations is applicable to other random privacy mechanisms as well. $\Delta_1 g = \max_{x,x' \in X} \|g(x) - g(x')\|_1$ measures the sensitivity of $g$ under the $\ell_1$ norm.

**A motivating example.** Consider an example shown in Fig. 1. For simplicity, we only consider an example with two people, Alice and Bob, and two data attributes, age and wage. Suppose we need to preserve $\epsilon$-LDP for each person with $\epsilon = 1$ and our task is to estimate the mean wage as closely as possible. A straightforward task-agnostic approach will directly expose both the two data attributes, and add Laplace noise with scale $b = 20$ to each attribute. However, a task-aware approach will expose only the wage attribute and add Laplace noise with scale $b = 10$. Both the approaches guarantee LDP under the same budget ($\epsilon = 1$), but the wage attribute given by the task-aware approach is less noisy, and we can expect that the corresponding estimated mean wage (i.e. the ultimate task objective) is close to the real value with a higher probability.

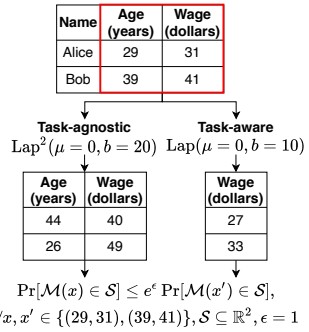

Figure 1: **Motivating example**. A task-aware approach (right) is more ideal than a task-agnostic approach (left) in terms of a mean wage estimation task, since the former perturbs the wage attribute with a smaller noise while guaranteeing the same LDP budget $\epsilon$.

In more complex scenarios, such as when each data attribute is not redundant but is valued differently in terms of the considered task or data attributes are dependent but not perfectly correlated, etc., the optimal solution will not be as straightforward as the given example and will be explored in the following sections.

## 3 PROBLEM FORMULATION

We now introduce the task-aware privacy preservation problem depicted in Fig. 2. Let $y = f(x) \in \mathbb{R}^m$ denote the task output associated with each ground truth data sample $x$, where $f$ represents the task function. To guarantee $\epsilon$-LDP for each data sample $x$, its true value should never be exposed to the task function. Instead, an estimate of $x$, denoted by $\hat{x}$, is used as the input to the task function with the corresponding task output $\hat{y} = f(\hat{x})$. The objective is to minimize the overall task loss $\mathcal{L} = \mathbb{E}[l(\hat{y}, y)]$ due to the difference between $\hat{x}$ and $x$, where $x$ follows distribution $\mathcal{D}_x$, and $l$ is a task loss function that captures the discrepancy between task output $\hat{y}$ and $y$, such as the common $\ell_2$ and cross-entropy loss.

More concretely, $x$ is first mapped to a latent representation $\phi \in \mathbb{R}^Z$ through an encoder function $\phi = g_e(x; \theta_e)$, where $\theta_e$ are a set of encoder parameters. $\phi$ is then perturbed by a Laplace noise vector $w \in \mathbb{R}^Z$. That is, $g_e$ is treated as the sensitive function $g$ in Eq.2. Next, $\hat{x}$ is reconstructed

from $\phi + w$ using a decoder function $\hat{x} = g_d(\phi + w; \theta_d)$ where $\theta_d$ are a set of decoder parameters. Note that in reality the encoder is deployed at the end of each individual user and in general has to be lightweight (e.g., linear or one hidden-layer neural network).

The optimal task-aware $\hat{x}$ minimizes $\mathcal{L}$ while preserving $\epsilon$-LDP. In other words, the task-aware privacy preservation problem aims to co-design encoder and decoder, i.e., find proper values for $Z$, $\theta_e$ and $\theta_d$, such that $\mathcal{L}$ is minimized and $\epsilon$-LDP is preserved. Formally,

$$\min_{Z, \theta_e, \theta_d} \quad \mathcal{L} = \mathbb{E}_{x,w}[l(\hat{y}, y)], \qquad (3)$$

$$\text{s.t.} \quad y = f(x), \qquad (4)$$

$$\hat{y} = f(g_d(g_e(x; \theta_e) + w; \theta_d)), \qquad (5)$$

$$x \sim \mathcal{D}_x, w \sim \text{Lap}^Z(0, \frac{\Delta_1 g_e}{\epsilon}). \qquad (6)$$

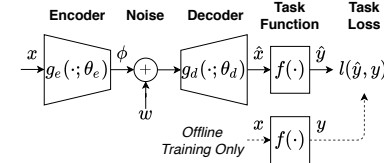

Figure 2: Overall architecture of the task-aware privacy preservation problem.

The difficulty of our task-aware LDP problem mainly comes from the discrepancy of the measurement of overall task loss $\mathcal{L}$, which depends on $\mathcal{D}_x$ and captures the *average* performance, and the mechanism of preserving LDP, which depends on $X$ and focuses only on the *worst-case* privacy guarantee.

**Benchmarks.** We now describe two natural approaches to preserve $\epsilon$-LDP. First, a **task-agnostic approach** adds noise directly to the normalized $x$[1]. For convenience we assume $x$ is already normalized, and we have $Z = n$ and $g_e(x) = x$. Second, a **privacy-agnostic approach** adds noise to $\phi$ obtained by considering the problem defined in Eq.3-6 with a pre-determined $Z \leq n$ and $w$ being a zero vector instead. That is, the privacy-preservation part is neglected when designing the encoder, and hence a proper $Z$ needs to be pre-determined or one would always conclude a larger $Z$ (under which more information can be delivered when noise is absent) is better. Both two benchmark approaches still need to determine the optimal decoder parameters $\theta_d$ for input $\phi + w$. Note that for the task-agnostic approach even though $g_e$ is an identity function, the corresponding optimal $g_d$ is usually not an identity function, as exemplified in Section A.7.1.

# 4 ANALYSIS

In this section, we solve the task-aware privacy preservation problem. Assuming a linear model and mean-squared error (MSE) task loss, we are able to find near-optimal analytical results, which shed clear insight on how to co-design an encoder and decoder. We then move to general settings and present a heuristic learning algorithm which demonstrates strong empirical performance.

## 4.1 LINEAR MODEL WITH MSE TASK LOSS

In this subsection, we consider a linear model with MSE task loss. More specifically, encoder function $g_e$, decoder function $g_d$, and task function $f$ are assumed to be linear functions in their corresponding inputs, and the loss function is $l = \|(\hat{y} - y)\|_2^2$. The task function $f$ can then be expressed as $f(x) = Kx$, where $K \in \mathbb{R}^{m \times n}$ is the task matrix.

**Practicality of the setting.** Linear transformation is a common encoding and decoding approach for many dimensionality-reduction techniques, such as PCA. And $\ell_2$ task loss is widely used in many application scenarios. For example, given $N$ samples of $x$, suppose we want to estimate the mean value of these samples in a few directions, given by task matrix $K$. Then the sum of the variance of these estimates by using $\hat{x}$ instead of $x$ will be $\frac{1}{N}\mathbb{E}_{x,w}[\|K(\hat{x}-x)\|_2^2]$, so $\mathcal{L} = \mathbb{E}_{x,w}[\|K(\hat{x}-x)\|_2^2] = \mathbb{E}_{x,w}[\|(\hat{y} - y)\|_2^2]$ is a natural objective function.

**Contents.** We first summarize key results. For our task-aware approach, the optimal decoder is first determined in Proposition 1, and then the optimal encoder and the corresponding optimal loss is

---

[1]We may either normalize each dimension independently when $x$ is a multi-variate random variable, or normalize all the dimensions jointly when $x$ is a uni-variate time-series.

formulated in Proposition 2-4 under Assumption 1. We then relax Assumption 1 and provide lower and upper bounds for the optimal loss in Theorem 1 and an approximate solution. All proofs are in the Appendix.

**Detailed analysis.** We start our analysis with a few definitions. First, without loss of generality, we assume the covariance matrix of $x - \mu_x$, i.e., $\mathbb{E}[(x - \mu_x)(x - \mu_x)^\top]$, is positive definite, where $\mu_x \in \mathbb{R}^n$ is the mean vector of $x$. This assumption guarantees $x$ cannot be linear transformed to a low-dimensional representation without losing information.

We then factorize $\mathbb{E}[(x - \mu_x)(x - \mu_x)^\top]$ into $LL^\top$ through Cholesky decomposition, where $L \in \mathbb{R}^{n \times n}$ is a lower triangular matrix with positive diagonal entries. For analytical convenience, we let $h = L^{-1}(x - \mu_x)$, which can be viewed as another representation of $x$, with mean $\mu_h = \mathbf{0}$ and covariance matrix $\Sigma_{hh} = I$. Let $\mathcal{D}_h$ denote the distribution of $h$, and $H = \{L^{-1}(x - \mu_x) | x \in X\}$ denote the compact set that contains all the possible values of $h \sim \mathcal{D}_h$. Since $K(\hat{x} - x) = P(\hat{h} - h)$, where $P = KL$, working with data representation $h$ with task matrix $P$ is equivalent to using $x$ and $K$. Considering zero-centered $h$ instead of original $x$ saves us from considering the constant terms in the linear encoder and decoder functions.

Let $E \in \mathbb{R}^{Z \times n}$ and $D \in \mathbb{R}^{n \times Z}$ denote the encoder and decoder matrix associated with $h$, i.e., $\phi = Eh$ and $\hat{h} = D(Eh + w)$. Without loss of generality, we let $Z \geq n$ and allow some rows of $E$ to be zero. Equivalently, based on the relationship between $x$ and $h$, we have $\phi = EL^{-1}(x - \mu_x)$ and $\hat{x} - \mu_x = LD(EL^{-1}(x - \mu_x) + w)$. We denote the covariance matrix of $w$ by $\Sigma_{ww}$, and it can be expressed as $\Sigma_{ww} = \sigma_w^2 I$, where $\sigma_w^2$ is the variance of the noise added to each dimension of $\phi$.

We first determine the optimal decoder $D$ that minimizes $\mathcal{L}$ for a given encoder $E$, which is given by the following proposition.

**Proposition 1** (Optimal decoder $D$ that minimizes $\mathcal{L}$). *An optimal decoder $D$ that minimizes $\mathcal{L}$ for a given encoder $E$ and $\sigma_w^2$ can be expressed as $D = E^\top(EE^\top + \sigma_w^2 I)^{-1}$, and corresponding $\mathcal{L}$ is*

$$\mathcal{L} = Tr(P^\top P) - Tr(P^\top P E^\top (EE^\top + \sigma_w^2 I)^{-1} E), \tag{7}$$

*where $Tr(\cdot)$ denotes the trace of a matrix.*

The next main step is to find an encoder $E$ that minimizes Eq.7. Since $\Delta_1 g_e = \max_{v,v' \in E(H)} \|v - v'\|_1$, where $E(H) = \{Eh | h \in H\}$ is the image of $H$ under linear transformation $E$, the design of encoder $E$ will affect $\Delta_1 g_e$ and therefore $\sigma_w^2$, and for different $H$'s the effect of $E$ is also different in general. So we need to carefully consider the relationship between $E$ and $H$.

When computing $\Delta_1 g_e$ we can actually use $H$'s convex hull $S$ instead of $H$ itself, according to the following lemma. This brings some analytical convenience in the further analysis.

**Lemma 1** (Convex hull preserves $\Delta_1 g_e$).

$$\Delta_1 g_e = \max_{v,v' \in E(S)} \|v - v'\|_1. \tag{8}$$

For encoder $E$, consider its singular value decomposition (SVD) instead: $U\Sigma V^\top$, where $U \in \mathbb{R}^{Z \times Z}$ and $V \in \mathbb{R}^{n \times n}$ are orthogonal matrices and $\Sigma \in \mathbb{R}^{Z \times n}$ is a rectangular diagonal matrix. And the singular values are denoted by $\sigma_1, \cdots, \sigma_n$ with $|\sigma_1| \geq \cdots \geq |\sigma_n|$. Then designing $E$ is equivalent to designing matrix $U, V$ and $\Sigma$. The geometric interpretation of applying transform $E = U\Sigma V^\top$ to set $S$ consists of three sub-transforms: 1) rotate $S$ by applying rotation matrix $V^\top$; 2) scale $V^\top(S)$ by applying scaling matrix $\Sigma$; 3) rotate $\Sigma V^\top(S)$ by applying rotation matrix $U$. In general, the choice of any of $U, \Sigma$, and $V$ will affect $\Delta_1 g_e$ and hence $\sigma_w^2$.

Our overall strategy is to first minimize the loss $\mathcal{L}$ and the sensitivity value $\Delta_1 g_e$ over $U$ and $V$. The resulting $U$ and $V$ only depend on $\Sigma$ under Assumption 1 (Propositions 2, 3). We then find the optimal $\Sigma$ that minimizes the $\mathcal{L}$ within a given privacy budget $\epsilon$ (Proposition 4). By the end, we relax Assumption 1 to discuss the quality of our solution (Theorem 1).

We start by noting that for two points within a compact set, they must lay on the boundary to have the maximum $\ell_1$ distance. We then make the following assumption in terms of $\partial S$, which is the boundary of $S$. It decouples the relationship between the choice of $V$ and the value of $\Delta_1 g_e$.

**Assumption 1** (Boundary $\partial S$ is a centered hypersphere). *The boundary $\partial S$ is a centered hypersphere of radius $r \geq 0$, which is expressed as $\{h \in \mathbb{R}^n | \|h\|_2^2 = r^2\}$.*

This is a strong assumption, but at the end of this subsection we will give a lower and upper bound of $\mathcal{L}$ for any possible $\partial S$ based on the results under this assumption. Since $\partial V(S) = \{h \in \mathbb{R}^n | \|h\|_2^2 = r^2\} = \partial S$ for any orthogonal $V$, this assumption gives us a nice property: the choice of $V$ doesn't affect $\Delta_1 g_e$ and $\sigma_w^2$. Based on the above assumption, we can safely consider the optimal design of $V$ that minimizes $\mathcal{L}$ when $\Sigma$ and $\sigma_w^2$ are given, which leads to the following proposition.

**Proposition 2** (Optimal rotation matrix $V$ that minimizes $\mathcal{L}$ under Assumption 1). *Suppose the eigen-decomposition of the Gram matrix $P^\top P$ is expressed as $P^\top P Q = Q \Lambda$, where $\Lambda = diag(\lambda_1, \cdots, \lambda_n) \in \mathbb{R}^{n \times n}$ is a diagonal matrix whose diagonal elements are eigenvalues with $\lambda_1 \geq \cdots \geq \lambda_n \geq 0$, and $Q \in \mathbb{R}^{n \times n}$ is an orthogonal matrix whose columns are corresponding normalized eigenvectors. Then, when $\Sigma$ and $\sigma_w^2$ are given, $\mathcal{L}$ is minimized for $V = Q$, any $Z \geq n$, and any orthogonal $U$. And the corresponding $\mathcal{L}$ can be expressed as:*

$$\mathcal{L} = \sum_{i=1}^{n} \lambda_i - \sum_{i=1}^{n} \lambda_i \frac{\sigma_i^2}{\sigma_i^2 + \sigma_w^2}. \tag{9}$$

It is clear that choosing a $Z > n$ brings no additional benefit. Hence we can only consider $Z = n$ for simplicity.

After the first two sub-transforms $\Sigma$ and $V^\top$, the boundary $\partial S = \{h \in \mathbb{R}^n | \|h\|_2^2 = r^2\}$ becomes $\partial \Sigma V^\top(S) = \{v \in \mathbb{R}^n | \sum_{i=1}^{n} v_i^2 / \sigma_i^2 = r^2\}$, which is a hyperellipsoid. We then have the following proposition that gives the optimal $U$ which minimizes $\Delta_1 g_e$.

**Proposition 3** (Optimal rotation matrix $U$ that minimizes $\Delta_1 g_e$ under Assumption 1). *For a given $\Sigma$, $U = I$ minimizes $\Delta_1 g_e$, and the corresponding minimum value is*

$$\Delta_1 g_e = 2r \sqrt{\sum_{i=1}^{n} \sigma_i^2}. \tag{10}$$

Next, we need to consider how to design the scaling matrix $\Sigma$, or equivalently, the values of $\sigma_1^2, \cdots, \sigma_n^2$, to minimize Eq.9, which is the only remaining piece. Clearly, for any given $\sigma_1^2, \cdots, \sigma_n^2$ if we increase them proportionally, $\sigma_w^2$ also needs to be increased proportionally to preserve the same $\epsilon$-LDP. So without loss of generality, we impose an additional constraint $\sum_{i=1}^{n} \sigma_i^2 = M$, where $M$ is a positive constant. And the following proposition gives the optimal choice of $\Sigma$ that minimizes $\mathcal{L}$ and preserves $\epsilon$-LDP with the Laplace mechanism.

**Proposition 4** (Optimal scaling matrix $\Sigma$ that minimizes $\mathcal{L}$ and preserves $\epsilon$-LDP with Laplace mechanism under Assumption 1). *The optimal choice of $\sigma_1^2, \cdots, \sigma_n^2$ that minimize $\mathcal{L}$ and preserve $\epsilon$-LDP with Laplace mechanism under constraint $\sum_{i=1}^{n} \sigma_i^2 = M$ is given by*

$$\sigma_i^2 = \begin{cases} M \cdot \left( \frac{\sqrt{\lambda_i}}{\sum_{i=1}^{Z'} \sqrt{\lambda_i}} (1 + Z' \cdot \frac{8r^2}{\epsilon^2}) - \frac{8r^2}{\epsilon^2} \right), & \forall i \in \{1, 2, \cdots, Z'\} \\ 0, & \forall i \in \{Z'+1, Z'+2, \cdots, n\} \end{cases} \tag{11}$$

*where $Z' \leq n$ is the largest integer such that:*

$$\frac{\sqrt{\lambda_{Z'}}}{\sum_{i=1}^{Z'} \sqrt{\lambda_i}} (1 + Z' \cdot \frac{8r^2}{\epsilon^2}) - \frac{8r^2}{\epsilon^2} > 0, \tag{12}$$

*and the corresponding $\mathcal{L}$ is*

$$\mathcal{L} = \frac{8r^2/\epsilon^2}{1 + Z' \cdot 8r^2/\epsilon^2} \left( \sum_{i=1}^{Z'} \sqrt{\lambda_i} \right)^2 + \sum_{i=Z'+1}^{n} \lambda_i. \tag{13}$$

For our task-aware approach, Proposition 1-4 complete the optimal encoder and decoder design that preserves $\epsilon$-LDP with the Laplace mechanism under Assumption 1.

**Validation of Task Loss from Theoretical Results (Fig. 3).** We now compare the performance of our task-aware approach and the benchmark approaches when $L = I$ and Assumption 1 holds. The derivations of the benchmark approaches are in Section A.7. We consider three different settings

which have $n = 4$ and $\lambda_1 = 4$ in common. And in setting 1, 2, and 3 we let $\lambda_2 = \lambda_3 = \lambda_4$ be 0, 1, 2 respectively. For the privacy-agnostic approach[2], we use a pre-determined $Z = 2$. Our observations are: 1) Compared to the task-agnostic approach, our task-aware approach achieves the largest improvement in setting 1, because $\lambda_{2:4} = 0$ implies that $x_{2:4}$ are purely redundant. We can even expect higher gain than setting 1 when we have larger $n$ and zero $\lambda_{2:n}$'s, and the gain will be zero if all the $\lambda_i$'s are equal. 2) The privacy-agnostic approach completely missed the information carried by $x_{3:4}$, which explains the improvement of our task-aware approach for small $r/\epsilon$ in setting 2 and 3. We can expect higher gain than setting 3 when we have larger $n$ and larger $\lambda_{2:n}$'s, and the gain will be zero if all the missed $x_i$'s correspond to zero $\lambda_i$'s and all the other $\lambda_i$'s are equal.

**Transition to general boundary $\partial S$.** Based on the results under Assumption 1, we can give a lower and upper bound of $\mathcal{L}^*$ for general $\partial S$, which is not necessarily a centered hypersphere.

**Theorem 1** (Lower and upper bound of $\mathcal{L}^*$ for general boundary $\partial S$ when $\epsilon$-LDP is preserved with Laplace mechanism). *Suppose $\partial S \subset \{h \in \mathbb{R}^n : r_{min}^2 \leq \|h\|_2^2 \leq r_{max}^2\}$. Then when $\epsilon$-LDP is preserved with the Laplace mechanism, the optimal $\mathcal{L}^*$ is bounded by:*

$$\mathcal{L}(r_{min}; \lambda_{1:n}, \epsilon) \leq \mathcal{L}^* \leq \mathcal{L}(r_{max}; \lambda_{1:n}, \epsilon) \quad (14)$$

*where $\mathcal{L}(r; \lambda_{1:n}, \epsilon)$ is the value of $\mathcal{L}$ determined by Eq.12 and 13 for given radius $r$, eigenvalues $\lambda_{1:n}$ and privacy budget $\epsilon$.*

Figure 3: Theoretical overall task loss $\mathcal{L}$ comparison when $L = I$ and Assumption 1 holds. We consider three different settings which have $n = 4$ and $\lambda_1 = 4$ in common. Difference between settings: 1) $\lambda_{2:4} = 0$; 2) $\lambda_{2:4} = 1$; 3) $\lambda_{2:4} = 2$. For the privacy-agnostic approach we use $Z = 2$.

Therefore, to preserve $\epsilon$-LDP with the Laplace mechanism, our task-aware solution for general $\partial S$ is: 1) First, find the smallest $r_{max}$ and the largest $r_{min}$ that bound $\partial S$; 2) Then assume $\partial S$ is $\{h \in \mathbb{R}^n | \|h\|_2^2 = r_{max}^2\}$, and choose the encoder $E$ and decoder $D$ based on Proposition 2-4. We don't use the corresponding $\sigma_w^2$ however, because it may guarantee a higher LDP than needed. 3) Next, compute $\sigma_w^2$ for real $\partial S$ under decoder $D$ and privacy budget $\epsilon$.

The associated loss for the task-aware approach is at most $\mathcal{L}(r_{max}; \lambda_{1:n}, \epsilon)$. Though in general not optimal, it differs from $\mathcal{L}^*$ by at most $\mathcal{L}(r_{max}; \lambda_{1:n}, \epsilon) - \mathcal{L}(r_{min}; \lambda_{1:n}, \epsilon)$. The difference is small when $\partial S$ is "nearly" a hypersphere, i.e., $r_{max} - r_{min} \approx 0$.

## 4.2 GENERAL SETTINGS

For more complex scenarios, it is challenging to give an analytical solution to the task-aware privacy preservation problem, especially when the encoder function $g_e$, decoder function $g_d$, and task function $f$ correspond to neural networks. Thus, we present a heuristic learning algorithm. (The benchmark algorithms are given in Section A.7.4 due to space limit.)

Algorithm 1 is our proposed task-aware algorithm for general settings. First, the privacy budget $\epsilon$ and the latent dimension $Z$ are required inputs for the algorithm. In general, $Z$ should be proper, i.e., it is neither too small (we can find a better solution by choosing a larger $Z$) nor too big (which introduces unnecessary complexity). In practice a practitioner may need to determine a proper $Z$ on a case-by-case basis (See Section A.9 of the Appendix for more details). Next, the algorithm adopts an alternating iteration approach, where in each epoch, we first update parameters $\theta_e$, $\theta_d$ by their corresponding negative gradients in line 3, and then recompute $\Delta_1 g_e$ and re-sample $w$ from $\text{Lap}^Z(0, \Delta_1 g_e/\epsilon)$ in line 4. Note that, in terms of encoder parameter $\theta_e$, instead of considering the gradient of $\mathcal{L}$, we add an $\ell_2$ regularization term $\eta\|\theta_e\|^2$ where $\eta$ is a positive constant. Therefore, we update $\theta_e$ with the negative gradient $-(\nabla_{\theta_e}\mathcal{L} + 2\eta\theta_e)$. Without regularization, the $\|\theta_e\|^2$ will grow to infinity since we can always achieve a smaller $\mathcal{L}$ by increasing the scale of $\phi$ proportionally. But it is not a direction we are looking for, since $\sigma_w^2$ will also increase proportionally to guarantee $\epsilon$-LDP.

---

[2] One can use another value of $Z$, under which the result may be slightly different but our task-aware approach will still outperform.

---

**Algorithm 1** Task-aware Algorithm for $\epsilon$-LDP Preservation in General Settings

---

**Require:** Privacy budget $\epsilon$ and $Z$
1: Initialize encoder/decoder parameters $\theta_e, \theta_d$ and noise vector $w$
2: **for** $\tau \in \{0, 1, \cdots, N_{\text{epochs}} - 1\}$ **do**
3:      Update $\theta_e$ and $\theta_d$ with $-(\nabla_{\theta_e}\mathcal{L} + 2\eta\theta_e)$ and $-\nabla_{\theta_d}\mathcal{L}$, respectively, by one or multiple steps
4:      Recompute $\Delta_1 g_e$, and re-sample $w$ from $\text{Lap}^Z(0, \Delta_1 g_e/\epsilon)$
5: **end for**
6: Return $\theta_e, \theta_d$ and $\Delta_1 g_e$

---

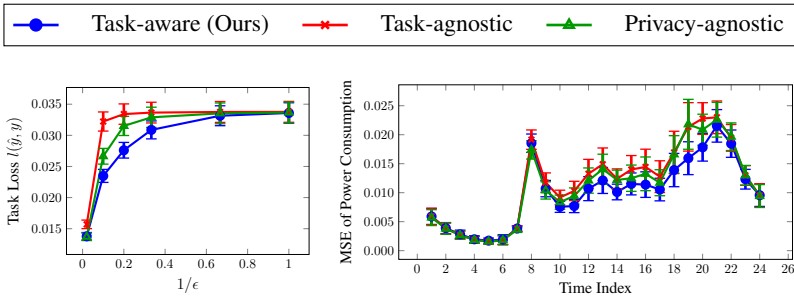

Figure 4: Results of Hourly Household Power Consumption. Left: Task loss $l(\hat{y}, y)$ under different LDP budgets. Right: MSE of power consumption for each hour, when $\epsilon = 5$. Our task-aware approach achieves a lower MSE for all the day-time hours.

## 5 EVALUATION

Our evaluation compares the performance of the proposed task-aware approach and the benchmark approaches. Three applications and corresponding datasets from the standard UCI Machine Learning Repository (Dua & Graff, 2017) are considered: mean estimation of hourly household power consumption, real estate valuation, and breast cancer detection. Configuration and training details are provided in the appendix due to space limitations.

**Mean Estimation of Hourly Household Power Consumption.** We first consider a mean estimation problem, based on measurements of individual household electric power consumption over four years (Hebrail & Berard, 2012). Each data sample $x \in \mathbb{R}^{24}$ is a time-series that contains the hourly household power consumption for one single day, and our objective is to estimate the mean of the hourly household power consumption for $N$ days. As discussed in Section 4.1, we can define the overall task loss in the following way:

$$\mathcal{L} = \mathbb{E}_{x \sim \mathcal{D}_x}[\|K(\hat{x} - x)\|_2^2] = \sum_{i=1}^{24} k_i^2 \mathbb{E}_{x \sim \mathcal{D}_x}[\|\hat{x}_i - x_i\|_2^2] \qquad (15)$$

where $K = \text{diag}(k_1, k_2, \cdots, k_{24})$ factors the importance of the mean estimation for each hour. In our experiment we set $k_i = 2$ for $i \in \{9, 10, \cdots, 20\}$ (i.e., day-time hours) and $k_i = 1$ for other $i$'s (i.e., night-time hours). And we adopt a linear encoder and decoder model. As the considered problem is based on a linear model with MSE task loss, we adopt the solutions developed in Section 4.1 for the three approaches (we choose $Z = 3$ for the privacy-agnostic approach[2]).

Fig. 4 shows our experimental results. First, on the left, we compare the task loss $l(\hat{y}, y)$ for the three approaches under different LDP budgets. For each approach, the overall task loss $\mathcal{L}$ decreases when a larger LDP budget $\epsilon$ is given. Besides, for a given LDP budget, our task-aware approach always outperforms the benchmark approaches on overall task loss $\mathcal{L}$, and the maximum improvements against the task-agnostic and privacy-agnostic approach are 22.9% ($\epsilon = 10$) and 11.7% ($\epsilon = 5$), respectively. Second, on the right, we select $\epsilon = 5$ and compare the MSE of power consumption for each hour. We see that our task-aware approach achieves a lower MSE for all the day-time hours, and a similar MSE for the night-time hours. This observation can be explained by three reasons: 1) We select a higher $k_i$ for the day-time hours, so our task-aware approach gives higher priority to

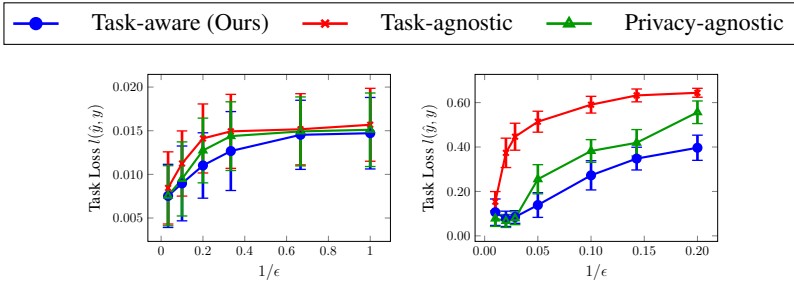

Figure 5: Task loss $l(\hat{y}, y)$ under different LDP budgets for real estate valuation (left) and breast cancer detection (right).

minimizing the loss for those dimensions in $x$; 2) Although $x$ has 24 dimensions, the variance in each dimension can be mostly explained by several common latent dimensions, so our task-aware approach still achieves a similar MSE for the night-time hours; 3) Our task-aware approach is able to adopt different scales to different latent dimensions according to their task relevance while the privacy-agnostic approach cannot.

Next, we consider a real estate valuation problem and a breast cancer detection problem. Both the problems are not based on the linear model with MSE task loss, so we use Algorithm 1 developed in Section 4.2 to solve them (we use $Z = 3$ for both our task-aware approach and the privacy-agnostic approach for fair comparison; and the performance of the task-aware approach under different $Z$'s can be found in Section A.9 of the Appendix).

**Real Estate Valuation.** For this problem, we use historical real estate valuation data collected from Taiwan (Yeh & Hsu, 2018), which contains 400+ instances. Here, $x \in \mathbb{R}^6$ contains 6 attributes that are highly related to the value of a house, including transaction date, house age, geographic coordinates, etc. And $y \in \mathbb{R}$ represents the valuation of a house. We first train a one-hidden-layer feedforward neural network regression model using the ground truth $x$ and $y$, to serve as our task function $f$. Then, we minimize the $\ell_2$ loss of $\hat{y}$ and $y$, based on a linear encoder and decoder model.

**Breast Cancer Detection.** For this problem, we use a well-known breast cancer diagnostic dataset (Street et al., 1993) from Wisconsin, which contains 500+ instances. Here, $x \in \mathbb{R}^{30}$ contains 30 attributes that measure 10 features of a cell nucleus. And $y$ is a binary variable that represents a diagnosis result (malignant or benign). We first train a one-hidden-layer feedforward neural network classification model using the ground truth $x$ and $y$, to serve as our task function $f$. Then we aim to minimize the cross-entropy loss of $\hat{y}$ and $y$, with encoder and decoder both being one-hidden-layer feedforward neural networks.

Fig. 5 shows the evaluation results. For both problems, we can see our task-aware approach nearly always outperforms the benchmark approaches on overall task loss $\mathcal{L}$ under different LDP budgets, which demonstrates the effectiveness of our proposed solution. The maximum improvements against the task-agnostic and privacy-agnostic approach are $21.9\%$ ($\epsilon = 5$) and $13.5\%$ ($\epsilon = 5$) for real estate valuation, and are $73.0\%$ ($\epsilon = 20$) and $45.6\%$ ($\epsilon = 20$) for breast cancer detection.

# 6 CONCLUSION AND FUTURE WORK

This paper provides a principled task-aware privacy preservation method to improve the privacy-utility trade-off for ML tasks that increasingly operate on rich, multi-dimensional user data. We gave an analytical near-optimal solution for a general linear encoder-decoder model and MSE task loss, and developed a heuristic learning algorithm for more general nonlinear settings. Our evaluation showed that our task-aware approach outperforms the benchmark approaches on overall task loss under various LDP budgets. Our future work will focus on the analysis of the task-aware privacy preservation problem for approximate LDP and other LDP mechanisms as well as multi-task learning.

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

# A APPENDIX

## A.1 PROOF OF PROPOSITION 1

*Proof.* We have

$$\mathcal{L} = \mathbb{E}[\|K(\hat{x} - x)\|_2^2] = \mathbb{E}[\|P(\hat{h} - h)\|_2^2] = \mathbb{E}[\|P((DE - I)h + Dw)\|_2^2] \tag{16}$$

$$= \mathbb{E}[\text{Tr}(P(DE - I)hh^\top(DE - I)^\top P^\top) + \text{Tr}(PDww^\top D^\top P^\top)] \tag{17}$$

$$= \text{Tr}(P(DE - I)(DE - I)^\top P^\top) + \text{Tr}(PD\Sigma_{ww}D^\top P^\top). \tag{18}$$

And we can verify that $D = E^\top(EE^\top + \sigma_w^2 I)^{-1}$ is a zero point of

$$\frac{\nabla \mathcal{L}}{\nabla D} = 2P^\top P((DE - I)E^\top + D \cdot \sigma_w^2 I). \tag{19}$$

Then we plug the expression of $D$ into Eq.18 and get Eq.7. □

## A.2 PROOF OF LEMMA 1

In this subsection we give the proof of Lemma 1. Here we treat $E$ as a mapping instead of a matrix, and its inverse mapping, which is also linear, is denoted by $E^{-1}$.

We first prove the following lemma.

**Lemma 2** (Convex hull after linear transformation). *$E(S)$ is the convex hull of $E(H)$.*

*Proof.* Since $H \subseteq S$, we have $E(H) \subseteq E(S)$. And for any $v, v' \in E(S)$ and $\delta \in [0, 1]$, we have $\delta h + (1 - \delta)h' \in S$, where $h \in E^{-1}(v) \cap S$ and $h' \in E^{-1}(v') \cap S$. Thus $\delta v + (1 - \delta)v' = E(\delta h + (1 - \delta)h') \in E(S)$. Therefore, $E(S)$ is a convex set containing $E(H)$.

If $E(S)$ is not the convex hull, then we can find a convex set $B$ such that $E(H) \subseteq B \subset E(S)$. Then we have $H \subseteq E^{-1}(B) \cap S \subset S$ (If $E^{-1}(B) \cap S = S$, we will have $B = E(E^{-1}(B) \cap S) = E(S)$, which is not true). Besides, for any $h, h' \in E^{-1}(B) \cap S$ and $\delta \in [0, 1]$, we have $\delta v + (1 - \delta)v' \in B$, where $v = E(h)$ and $v' = E(h')$. Thus $\delta h + (1 - \delta)h' \in E^{-1}(\delta v + (1 - \delta)v') \cap S \in E^{-1}(B) \cap S$. Therefore, $E^{-1}(B) \cap S$ is a convex set containing $H$. This is contradictory to the fact that $S$ is the convex hull of $H$. So $E(S)$ must be the convex hull of $E(H)$. □

Now we can proceed to the proof of Lemma 1.

*Proof.* Notice that $\Delta_1 g_e = \max_{v, v' \in E(H)} \|v - v'\|_1$, so our target is to prove

$$\max_{v, v' \in E(S)} \|v - v'\|_1 = \max_{v, v' \in E(H)} \|v - v'\|_1. \tag{20}$$

First, since $E(H) \subseteq E(S)$, we have $\max_{v, v' \in E(S)} \|v - v'\|_1 \geq \max_{v, v' \in E(H)} \|v - v'\|_1$.

Next, suppose $v_1, v_2$ are the two points in $E(S)$ such that the $\ell_1$ distance between them is the largest. Since $E(S)$ is the convex hull of $E(H)$, we can express $v_i$ as ($\forall i \in \{1, 2\}$):

$$v_i = \sum_{j=1}^{p(i)} t_{i,j} \tilde{v}_{i,j}, \tag{21}$$

$$\text{s.t.} \quad \tilde{v}_{i,j} \in E(H), \quad \forall j \in \{1, 2, \cdots, p(i)\}, \tag{22}$$

$$\sum_{j=1}^{p(i)} t_{i,j} = 1, \text{ and } t_{i,j} \geq 0, \quad \forall j \in \{1, 2, \cdots, p(i)\}, \tag{23}$$

where $p(i)$ is an positive integer, $\forall i \in \{1, 2\}$.

For convenience, we let $A = \max_{v, v' \in \{\tilde{v}_{i,j} | \forall i,j\}} \|v - v'\|_1$. Clearly, $A \leq \max_{v, v' \in E(H)} \|v - v'\|_1$.

Notice that $v_1 - v_2$ can be expressed as the linear combination of $\tilde{v}_{1,j} - \tilde{v}_{2,q}$, $\forall j \in \{1, 2, \cdots, p(1)\}, q \in \{1, 2, \cdots, p(2)\}$. That is, $\exists \gamma_{j,q} \geq 0$ such that:

$$v_1 - v_2 = \sum_{j=1}^{p(1)} \sum_{q=1}^{p(2)} \gamma_{j,q}(\tilde{v}_{1,j} - \tilde{v}_{2,q}), \quad \text{s.t.} \sum_{j=1}^{p(1)} \sum_{q=1}^{p(2)} \gamma_{j,q} = 1. \tag{24}$$

Then we have

$$\|v_1 - v_2\|_1 = \|\sum_{j=1}^{p(1)} \sum_{q=1}^{p(2)} \gamma_{j,q}(\tilde{v}_{1,j} - \tilde{v}_{2,q})\|_1 \tag{25}$$

$$\leq \sum_{j=1}^{p(1)} \sum_{q=1}^{p(2)} \|\gamma_{j,q}(\tilde{v}_{1,j} - \tilde{v}_{2,q})\|_1 = \sum_{j=1}^{p(1)} \sum_{q=1}^{p(2)} \gamma_{j,q}\|(\tilde{v}_{1,j} - \tilde{v}_{2,q})\|_1 \tag{26}$$

$$\leq \sum_{j=1}^{p(1)} \sum_{q=1}^{p(2)} \gamma_{j,q} A = A \leq \max_{v,v' \in E(H)} \|v - v'\|_1. \tag{27}$$

So we also have $\max_{v,v' \in E(S)} \|v - v'\|_1 \leq \max_{v,v' \in E(H)} \|v - v'\|_1$.

Thus $\Delta_1 g_e = \max_{v,v' \in E(H)} \|v - v'\|_1 = \max_{v,v' \in E(S)} \|v - v'\|_1$. □

### A.3 PROOF OF PROPOSITION 2

*Proof.* We have $\text{Tr}(P^\top P) = \sum_{i=1}^n \lambda_i$, which is a fixed number. Therefore we focus on maximizing the second trace term in Eq.7. For any $Z \geq n$ we have

$$EE^\top + \sigma_w^2 I = U\Sigma\Sigma^\top U^\top + \sigma_w^2 I = U\text{diag}(\sigma_1^2 + \sigma_w^2, \cdots, \sigma_n^2 + \sigma_w^2, \underbrace{\sigma_w^2, \cdots, \sigma_w^2}_{Z-n \text{ in total}})U^\top. \tag{28}$$

Thus we have for any orthogonal $U$:

$$E^\top(EE^\top + \sigma_w^2 I)^{-1}E \tag{29}$$

$$= V\Sigma^\top U^\top \cdot U\text{diag}(\frac{1}{\sigma_1^2 + \sigma_w^2}, \cdots, \frac{1}{\sigma_n^2 + \sigma_w^2}, \frac{1}{\sigma_w^2}, \cdots, \frac{1}{\sigma_w^2})U^\top \cdot U\Sigma V^\top \tag{30}$$

$$= V\Sigma^\top \text{diag}(\frac{1}{\sigma_1^2 + \sigma_w^2}, \cdots, \frac{1}{\sigma_n^2 + \sigma_w^2}, \frac{1}{\sigma_w^2}, \cdots, \frac{1}{\sigma_w^2})\Sigma V^\top \tag{31}$$

$$= V\text{diag}(\frac{\sigma_1^2}{\sigma_1^2 + \sigma_w^2}, \cdots, \frac{\sigma_n^2}{\sigma_n^2 + \sigma_w^2})V^\top. \tag{32}$$

So $E^\top(EE^\top + \sigma_w^2 I)^{-1}E$ is a positive semi-definite matrix with eigen-values $\frac{\sigma_1^2}{\sigma_1^2 + \sigma_w^2} \geq \cdots \geq \frac{\sigma_n^2}{\sigma_n^2 + \sigma_w^2} \geq 0$. Then by Ruhe's trace inequality (Ruhe, 1970) (a corollary of Von Neumanns trace inequality (Von Neumann, 1937)):

$$\text{Tr}(P^\top P E^\top (EE^\top + \sigma_w^2 I)^{-1}E) \leq \sum_{i=1}^n \lambda_i \frac{\sigma_i^2}{\sigma_i^2 + \sigma_w^2} \tag{33}$$

and when $V = Q$ the equality holds. So we have Eq.9 for $V = Q$, any $Z \geq n$ and any orthogonal $U$. □

### A.4 PROOF OF PROPOSITION 3

In this subsection we give the proof of Proposition 3. For convenience we let $a_i = r|\sigma_i|, \forall i \in \{1, 2, \cdots, n\}$, and then the hyperellipsoid $\{v \in \mathbb{R}^n | \sum_{i=1}^n v_i^2/\sigma_i^2 = r^2\}$ can be written as $\{v \in \mathbb{R}^n | \sum_{i=1}^n v_i^2/a_i^2 = 1\}$, which is the standard expression. And we also have $a_1 \geq a_2 \geq \cdots \geq a_n$.

Before proving Proposition 3, we first give two lemmas related to the properties of a hyperellipsoid.

**Lemma 3** (Tangent hyperplane of hyperellipsoid). *Any tangent hyperplane of hyperellipsoid $\{v \in \mathbb{R}^n | \sum_{i=1}^n v_i^2/a_i^2 = 1\}$ can be expressed as:*

$$\sum_{j=1}^n u_j v_j = \sqrt{\sum_{j=1}^n a_j^2 u_j^2}, \tag{34}$$

*where $u_1, u_2, \cdots, u_n$ are the coefficients.*

*Proof.* It can be easily verified that point $\tilde{v} \in \mathbb{R}^n$ such that

$$\tilde{v}_j = \frac{a_j^2 u_j}{\sqrt{\sum_{q=1}^n a_q^2 u_q^2}}, \quad j \in \{1, 2, \cdots, n\}, \tag{35}$$

is located on the hyperellipsoid. And by adjusting the values of $u_1, u_2, \cdots, u_n$ we can express any point on the hyperellipsoid with Eq.35. Moreover, the tangent hyperplane for point of tangency $\tilde{v}$ can be expressed as

$$\sum_{j=1}^n \frac{\tilde{v}_j v_j}{a_j^2} = 1. \tag{36}$$

Plugging Eq.35 into Eq.36 we get Eq.34. □

**Lemma 4** (Locus of the vertices of circumscribed orthotope). *The vertices of any orthotope that circumscribes hyperellipsoid $\{v \in \mathbb{R}^n | \sum_{i=1}^n v_i^2/a_i^2 = 1\}$ is on hypersphere $\{v \in \mathbb{R}^n | \sum_{i=1}^n v_i^2 = \sum_{i=1}^n a_i^2\}$.*

*Proof.* When $n = 2$, the hypersphere reduces to a circle that is well-known as the orthoptic circle of an ellipse (Casey, 1893). We hereby generalize this result to any $n$.

A vertex of a circumscribed orthotope can be viewed as the intersection of $n$ tangent hyperplanes. According to Lemma 3, we can express them as:

$$\sum_{j=1}^n u_{i,j} v_j = \sqrt{\sum_{j=1}^n a_j^2 u_{i,j}^2}, \quad \forall i \in \{1, 2, \cdots, n\}, \tag{37}$$

where $i$ is the index of the $i$-th hyperplane. Here we let $\sum_{j=1}^n u_{i,j}^2 = 1, \forall j \in \{1, 2, \cdots, n\}$. Besides, these hyperplanes are perpendicular to each other, so the coefficients also satisfy $\sum_{i=1}^n u_{i,j} u_{i,k} = 0$, $\forall k \neq j$. So if we let $\Omega \in \mathbb{R}^{n \times n}$ be a matrix with $u_{i,j}$ on the $i$-th row and $j$-th column, $\forall i, j$, then $\Omega$ is an orthogonal matrix, and we further have $\sum_{i=1}^n u_{i,j}^2 = 1, \forall i \in \{1, 2, \cdots, n\}$.

Thus the considered vertex satisfies Eq.37, $\forall i \in \{1, 2, \cdots, n\}$, which implies it also satisfies:

$$\sum_{i=1}^n (\sum_{j=1}^n u_{i,j} v_j)^2 = \sum_{i=1}^n \sum_{j=1}^n a_j^2 u_{i,j}^2. \tag{38}$$

For the left hand side, we have

$$\sum_{i=1}^n (\sum_{j=1}^n u_{i,j} v_j)^2 = \sum_{i=1}^n \sum_{j=1}^n u_{i,j}^2 v_j^2 + 2 \sum_{i=1}^n \sum_{j \neq p} u_{i,j} u_{i,k} v_j v_p \tag{39}$$

$$= \sum_{j=1}^n (\sum_{i=1}^n u_{i,j}^2) v_j^2 + 2 \sum_{j \neq p} (\sum_{i=1}^n u_{i,j} u_{i,k}) v_j v_p = \sum_{j=1}^n v_j^2. \tag{40}$$

And for the right hand side,

$$\sum_{i=1}^n \sum_{j=1}^n a_j^2 u_{i,j}^2 = \sum_{j=1}^n (\sum_{i=1}^n u_{i,j}^2) a_j^2 = \sum_{j=1}^n a_j^2. \tag{41}$$

Thus the vertex is on hypersphere $\{v \in \mathbb{R}^n | \sum_{i=1}^n v_i^2 = \sum_{i=1}^n a_i^2\}$. □

Since the equation of a centered hypersphere after rotation remains unchanged, we have the following corollary.

**Corollary 1** (Locus of the vertices of circumscribed orthotope after rotation). *The vertices of any orthotope that circumscribes hyperellipsoid $\{v \in \mathbb{R}^n | \sum_{i=1}^n v_i^2 / a_i^2 = 1\}$ after rotation is on hypersphere $\{v \in \mathbb{R}^n | \sum_{i=1}^n v_i^2 = \sum_{i=1}^n a_i^2\}$.*

Now we can proceed to the proof of Proposition 3.

*Proof.* We first consider the case when $n$ is a power of 2.

For the rotated hyperellipsoid, we consider whether there's any point $\tilde{v} \in \mathbb{R}^n$ s.t. $\|\tilde{v}\|_1 \geq \sqrt{\sum_{j=1}^n a_j^2}$. Suppose we couldn't find such a point. Then consider any tangent hyperplane whose normal vector has the following form: $(u_1, u_2, \cdots, u_n)$, s.t. $u_i = \pm 1, \forall i \in \{1, 2, \cdots, n\}$. It can be expressed as:

$$\sum_{j=1}^n u_j v_j = W(u_1, u_2, \cdots, u_n), \tag{42}$$

where $W : (u_1, u_2, \cdots, u_n) \mapsto \mathbb{R}$ maps $(u_1, u_2, \cdots, u_n)$ to a corresponding constant. And we have $W(u_1, u_2, \cdots, u_n) < \sqrt{\sum_{j=1}^n a_j^2}$.

Since $n$ is a power of 2, we can find a Hadamard matrix $\Omega$ in $\mathbb{R}^{n \times n}$ whose elements are either 1 or -1, such that $\Omega \Omega^\top = nI$. We let $u_{i,j}$ be the element of $R$ on the $i$-th row and $j$-th column, $\forall i, j$, and consider the intersection of the following $n$ tangent hyperplanes:

$$\sum_{j=1}^n u_{i,j} v_j = W(u_{i,1}, u_{i,2}, \cdots, u_{i,n}), \quad \forall i \in \{1, 2, \cdots, n\}, \tag{43}$$

whose intersection point must satisfy:

$$\sum_{i=1}^n (\sum_{j=1}^n u_{i,j} v_j)^2 = \sum_{i=1}^n \sum_{j=1}^n (W(u_{i,1}, u_{i,2}, \cdots, u_{i,n}))^2. \tag{44}$$

Similar to the proof of Lemma 4, we can easily prove the left hand side equals $n \sum_{j=1}^n v_j^2$, and the right hand side is strictly less than $n \sum_{j=1}^n a_j^2$. Thus the intersection point doesn't locate on the hypersphere $\{v \in \mathbb{R}^n | \sum_{i=1}^n v_i^2 = \sum_{i=1}^n a_i^2\}$. But since the considered $n$ hyperplanes are also the surfaces of a circumscribed orthotope, the intersection point is hence a vertex and must be on the hypersphere $\{v \in \mathbb{R}^n | \sum_{i=1}^n v_i^2 = \sum_{i=1}^n a_i^2\}$, according to Corollary 1. This contradiction means that, the assumption that we cannot find a point $\tilde{v} \in \mathbb{R}^n$ s.t. $\|\tilde{v}\|_1 \geq \sqrt{\sum_{j=1}^n a_j^2}$ is false.

Thus the point $\tilde{v} \in \mathbb{R}^n$ s.t. $\|\tilde{v}\|_1 \geq \sqrt{\sum_{j=1}^n a_j^2}$ exists, and the $\ell_1$ distance between $\tilde{v}$ and $-\tilde{v}$ (which both lie on the hyperepllisoid) is $2\sqrt{\sum_{j=1}^n a_j^2}$. Thus we have $\Delta_1 g_e \geq 2\sqrt{\sum_{j=1}^n a_j^2}$.

For $U = I$, which means we don't actually rotate the ellipsoid $\{v \in \mathbb{R}^n | \sum_{i=1}^n v_i^2 / a_i^2 = 1\}$, we have for any point on the ellipsoid

$$(\sum_{i=1}^n |v_i|)^2 = (\sum_{i=1}^n a_i \cdot \frac{|v_i|}{a_i})^2 \leq (\sum_{i=1}^n a_i^2)(\sum_{i=1}^n \frac{v_i^2}{a_i^2}) = \sum_{i=1}^n a_i^2, \tag{45}$$

according to the Cauchy-Schwartz inequality. This implies any point $v$ on the hyperellipsoid has $\|v\|_1 \leq \sqrt{\sum_{j=1}^n a_j^2}$. So $\Delta_1 g_e \leq 2\sqrt{\sum_{j=1}^n a_j^2}$. Combined with the result in the above paragraph we know $\Delta_1 g_e = 2\sqrt{\sum_{j=1}^n a_j^2}$ for $U = I$.

Thus the proposition is proved for $n$ being a power of 2. For other $n$'s, we can treat the considered hyperellipsoid as a degenerated hyperellipsoid in space $\mathbb{R}^{\tilde{n}}$, where $\tilde{n}$ is the smallest power of 2 such that $\tilde{n} > n$. This implies the proposition still holds.

Therefore, the proposition is true for any $n$. □

## A.5 Proof of Proposition 4

*Proof.* First, to preserve $\epsilon$-LDP with Laplace mechanism, the minimum $\sigma_w^2$ required is:

$$\sigma_w^2 = 2 \cdot \frac{(\Delta_1 g_e)^2}{\epsilon^2} = 2 \cdot \frac{4r^2 \cdot \sum_{i=1}^n \sigma_i^2}{\epsilon^2} = \frac{8r^2 M}{\epsilon^2}, \tag{46}$$

based on Proposition 3 and constraint $\sum_{i=1}^n \sigma_i^2 = M$.

Next we need to determine $\sigma_1^2, \cdots, \sigma_n^2$. The considered problem is an optimization problem which aims at minimizing $\mathcal{L}$ in Eq.9 under constraint $\sum_{i=1}^n \sigma_i^2 = M$ and $\sigma_i^2 \geq 0$ (here we view $\sigma_i^2$ instead of $\sigma_i$ as the decision variable). Note that though in Eq.9 we also have $\sigma_1^2 \geq \sigma_n^2$, we don't need to explicitly consider this constraint, because minimizing $\mathcal{L}$ will implicitly guarantee that, according to the rearrangement inequality. This problem can be solved by KarushKuhnTucker (KKT) approach, with the following Lagrangian function:

$$F(\sigma_1^2, \cdots, \sigma_n^2, \alpha_1, \cdots, \alpha_n, \beta) = \sum_{i=1}^n \lambda_i - \sum_{i=1}^n \lambda_i \frac{\sigma_i^2}{\sigma_i^2 + \sigma_w^2} + \sum_{i=1}^n \alpha_i(-\sigma_i^2) + \beta(\sum_{i=1}^n \sigma_i^2 - M), \tag{47}$$

where $\alpha_1, \cdots, \alpha_n$ and $\beta$ are Lagrangian multipliers. We know that the solution will automatically guarantee $\sigma_1^2 \geq \cdots \geq \sigma_n^2$, so we can safely assume there exists $Z' \leq n$ such that $\alpha_i = 0$ for $i \leq Z'$, and $\alpha_i > 0$ for $i > Z'$. Then for $i > Z'$ we have $\sigma_i^2 = 0$, and for $i \leq Z'$ we have

$$\frac{\nabla F}{\nabla \sigma_i^2} = -\frac{\sigma_w^2}{(\sigma_i^2 + \sigma_w^2)^2} \lambda_i = \beta. \tag{48}$$

Combining with $\sum_{i=1}^{Z'} \sigma_i^2 = M$ and Eq.46 we eventually get Eq.11. Enforcing $\sigma_{Z'}^2 > 0$ we get Eq.12. And plugging Eq.11 into Eq.9 we get Eq.13. $\qquad \square$

## A.6 Proof of Theorem 1

In this subsection we give the proof of Theorem 1.

We start with two definitions: $\mathcal{L}^*(\partial S; P, \epsilon)$ denotes the optimal loss for any boundary $\partial S$, task matrix $P$ that preserves $\epsilon$-LDP with Laplace mechanism; $R(r) = \{h \in \mathbb{R}^n | \|h\|_2^2 = r^2\}$ is the centered hypersphere with radius $r$.

Next we give the following lemma.

**Lemma 5** (Invariance of the optimal loss after scaling)**.**

$$\mathcal{L}^*(\partial S; P, \epsilon) = \mathcal{L}^*(\partial \rho(S); P, \rho\epsilon), \tag{49}$$

*where $\rho > 0$ is a scalar and $\rho(S) = \{\rho h | h \in S\}$.*

*Proof.* We only need to consider fixed task matrix $P$. For any encoder $E$, decoder $D$ and noise variance $\sigma_w^2$, if they preserve $\epsilon$-LDP with Laplace mechanism for boundary $\partial S$, then we have

$$\sigma_w \geq \sqrt{2} \cdot \frac{\Delta_1 g_e}{\epsilon} = \sqrt{2} \cdot \frac{\Delta_1 \rho g_e}{\rho\epsilon}, \tag{50}$$

where

$$\Delta_1 \rho g_e = \rho \max_{h,h' \in S} \|Eh - Eh'\|_1 = \max_{v,v' \in \rho(S)} \|Ev - Ev'\|_1. \tag{51}$$

Thus we know encoder $E$, decoder $D$ and noise variance $\sigma_w^2$ also preserve $\rho\epsilon$-LDP with Laplace mechanism for boundary $\partial \rho(S)$. And the reverse also holds true.

So we must have $\mathcal{L}^*(\partial S; P, \epsilon) = \mathcal{L}^*(\partial \rho(S); P, \rho\epsilon)$. $\qquad \square$

Now we can proceed to the proof of Theorem 1.

*Proof.* We first construct a distribution $\mathcal{D}_{h'}$, s.t. points drawn from $\mathcal{D}_{h'}$ are uniformly distributed on $R(2^{n-1})$. Then $h' \sim \mathcal{D}_{h'}$ satisfies $\Sigma_{h'h'} = I$. And we also have $\partial S' = R(2^{n-1})$, where $S'$ is the convex hull of $H'$ ($H'$ is the domain of $h' \sim \mathcal{D}_{h'}$). According to Proposition 4, for $h' \sim \mathcal{D}_{h'}$ we have $\mathcal{L}^*(R(2^{n-1}); P, \rho\epsilon) = \mathcal{L}(2^{n-1}; \lambda_{1:n}, \rho\epsilon), \forall \rho, \epsilon > 0$.

We next consider scalar $\rho = 2^{n-1}/r_{\min}$ and the optimal loss $\mathcal{L}^*(\partial\rho(S); P, \rho\epsilon)$. For any encoder $E$, decoder $D$ and noise variance $\sigma_w^2$, if they preserve $\rho\epsilon$-LDP to boundary $\partial\rho(S)$, then they preserve at least $\rho\epsilon$-LDP to boundary $R(2^{n-1})$, since $R(2^{n-1}) \subset \rho(S)$. Thus we have $\mathcal{L}^*(\partial\rho(S); P, \rho\epsilon) \geq \mathcal{L}^*(R(2^{n-1}); P, \rho\epsilon)$.

This further implies:

$$\mathcal{L}^*(\partial S; P, \epsilon) = \mathcal{L}^*(\partial\rho(S); P, \rho\epsilon) \tag{52}$$

$$\geq \mathcal{L}^*(R(2^{n-1}); P, \rho\epsilon) = \mathcal{L}(2^{n-1}; \lambda_{1:n}, \rho\epsilon) = \mathcal{L}(r_{\min}; \lambda_{1:n}, \epsilon). \tag{53}$$

So the lower bound of Theorem 1 is proved. The upper bound can be proved in the same way. □

### A.7 BENCHMARKS

#### A.7.1 TASK LOSS FOR TASK-AGNOSTIC APPROACH

One can obtain the resultant optimal $\mathcal{L}$ for the task-agnostic approach by letting $E = L$ and using Eq.7 as stated in Proposition 1. It is worth noting that the associated decoder $D = L^\top(LL^\top + \sigma_w^2 I)^{-1}$ is not an identity matrix in general.

**Corollary 2** (Optimal $\mathcal{L}$ for the task-agnostic approach that preserves $\epsilon$-LDP). *For the task-agnostic approach, the optimal $\mathcal{L}$ that preserves $\epsilon$-LDP is*

$$\mathcal{L} = Tr(P^\top P) - Tr(P^\top P L^\top(LL^\top + \sigma_w^2 I)^{-1}L), \tag{54}$$

*where $\sigma_w^2 = 2(\Delta_1 g_e)^2/\epsilon^2$ with $g_e(x) = x$.*

#### A.7.2 TASK LOSS FOR PRIVACY-AGNOSTIC APPROACH

Through similar analysis as Proposition 2, one can obtain the resultant optimal $\mathcal{L}$ for the privacy-agnostic approach, which has a pre-determined $Z \leq n$.

**Corollary 3** (Optimal $\mathcal{L}$ for the privacy-agnostic approach that preserves $\epsilon$-LDP). *For the privacy-agnostic approach with a pre-determined $Z \leq n$, the optimal $\mathcal{L}$ that preserves $\epsilon$-LDP is*

$$\mathcal{L} = \sum_{i=1}^{Z} \lambda_i \frac{\sigma_w^2}{\sigma_i^2 + \sigma_w^2} + \sum_{i=Z+1}^{n} \lambda_i. \tag{55}$$

*where $\sigma_w^2 = 2(\Delta_1 g_e)^2/\epsilon^2$.*

*Proof.* For privacy-agnostic approach, we have $Z \leq n$. Since for encoder we need to select the top-$Z$ principal components, we have $V = Q$. Similar to the proof of Proposition 2, for a given $Z \leq n$, we have

$$EE^\top + \sigma_w^2 I = U\Sigma\Sigma^\top U^\top + \sigma_w^2 I = U\text{diag}(\sigma_1^2 + \sigma_w^2, \cdots, \sigma_Z^2 + \sigma_w^2)U^\top, \tag{56}$$

and through similar derivations we eventually get

$$Tr(P^\top P E^\top(EE^\top + \sigma_w^2 I)^{-1}E) = \sum_{i=1}^{Z} \lambda_i \frac{\sigma_i^2}{\sigma_i^2 + \sigma_w^2}. \tag{57}$$

Combined with Eq. 7 we get Eq. 55. □

### A.7.3 Task loss for benchmark approaches when $L = I$ and Assumption 1 holds

When $L = I$ and Assumption 1 holds, for the task-agnostic approach, according to Eq. 54, we have

$$\mathcal{L} = \frac{\sigma_w^2}{1 + \sigma_w^2} \cdot \text{Tr}(P^\top P) = \frac{n \cdot 8r^2/\epsilon^2}{1 + n \cdot 8r^2/\epsilon^2} \sum_{i=1}^{n} \lambda_i. \tag{58}$$

For the privacy-agnostic approach, according to Eq. 55 and Eq. 10, and assuming we have equal $\sigma_i$'s and minimum $\delta_1 g_e$, we get

$$\mathcal{L} = \frac{Z \cdot 8r^2/\epsilon^2}{1 + Z \cdot 8r^2/\epsilon^2} \sum_{i=1}^{Z} \lambda_i + \sum_{i=Z+1}^{n} \lambda_i, \tag{59}$$

where the value of $Z$ is pre-determined.

### A.7.4 Benchmark algorithms under general settings

For the privacy-agnostic approach, we first train the encoder and decoder without considering privacy preservation by updating $\theta_e$ and $\theta_d$ with $-\nabla_{\theta_e}\mathcal{L}$ and $-\nabla_{\theta_d}\mathcal{L}$, respectively. Next, we fix encoder parameters $\theta_e$ and train the decoder with input $\phi + w$ (a modification of Algorithm 1 line 3-4). The task-agnostic approach trains the decoder in the same way, but fixes $g_e$ to an identity mapping function.

### A.8 Configuration and Training Details of the Evaluation

Table 1: Evaluation Details

| Application | Num of Samples | Train/Test Split | Training Epochs | Runtime |
|---|---|---|---|---|
| Household Power | 1417 | 0.7/0.3 | NA | < 1 min |
| Real Estate | 414 | 0.7/0.3 | 2000 | < 2 hrs |
| Breast Cancer | 569 | 0.7/0.3 | 2000 | < 2 hrs |

Our evaluation runs on a personal laptop with 2.7 GHz Intel Core I5 processor and 8-GB 1867 MHz DDR3 memory. Our code is based on Pytorch. We use the Adam optimizer and learning rate $10^{-3}$ for all the applications. The number of samples, train/test split, training epochs, and resulting runtime are summarized in Table 1. (Note that the evaluation for hourly household power consumption is based on the theoretical solutions, so "training epochs", which is associated with the gradient-based method, doesn't apply.) **We also provide all our documented code as supplementary material and will make it publicly-available after the review process**. All three datasets cited in the evaluation are publicly-available from the standard UCI Machine Learning Repository (Dua & Graff, 2017) and anonymized using standard practices. The individual dataset licenses were not available.

For task function $f$, we use a one-hidden-layer feedforward neural network with input size $n$, hidden size $1.5n$ and output size 1 in both the real estate valuation and breast cancer detection experiments. The activation function used by the hidden layer and output layer is a Rectified Linear Unit (ReLU). In our experiments, we find that the chosen network architecture is good enough to yield near-zero loss with ground truth $x$ and $y$, and to **avoid overfitting** we don't choose a deep neural network. For example, we did not see any task improvement using a two layer network etc.

For the encoder/decoder, we use a one-layer neural network (linear model) with input and output size $n$ in the real estate valuation experiment. For this experiment, a linear encoder/decoder model is already enough to provide good performance (see the next subsection for further details). We use one-hidden-layer feedforward neural network with input size $n$, hidden size $n$ and output size $n$ in the breast cancer detection experiment. The activation functions used by the hidden layer and output layer are a logistic and identity function, respectively.

For the heuristic learning algorithm developed in Section 4.2, we set $\eta = 0.2$ and $\eta = 0.001$ in real estate valuation and breast cancer detection experiments, respectively; and in both experiments, for each epoch we update $\theta_e$ and $\theta_d$ by 15 steps.

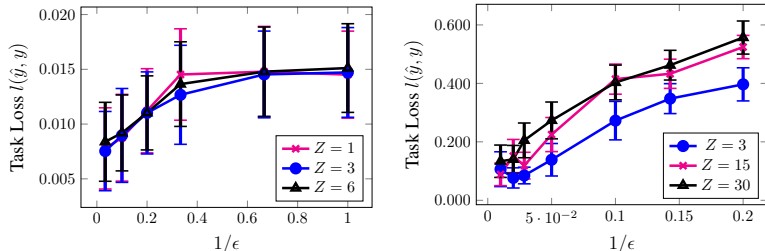

Figure 6: Task loss $l(\hat{y}, y)$ of our task-aware approach under different $Z$'s for real estate valuation (left) and breast cancer detection (right).

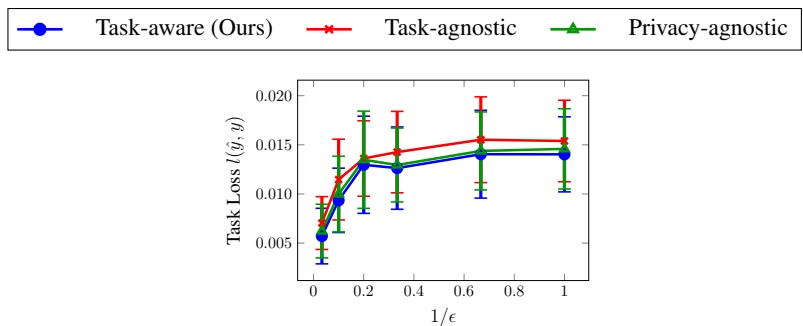

Figure 7: Task loss $l(\hat{y}, y)$ under different LDP budgets for real estate valuation, when a one-hidden-layer feedforward neural network is adopted for both encoder and decoder.

### A.9 A Proper $Z$ should be Determined on a Case-by-case Basis

As mentioned in Section 4.2, a practitioner may need to determine a proper $Z$ on a case-by-case basis for our task-aware approach. Fig. 6 illustrates the performance of our task-aware approach under different $Z$'s for real estate valuation and breast cancer detection experiments. We can observe that in both two experiments, we obtained the best performance on average when $Z = 3$, i.e., $n/2$ for real estate valuation and $n/10$ for breast cancer detection[3].

### A.10 Non-Linear Encoder and Decoder Model in Real Estate Valuation Experiment

Theoretically, increasing the model complexity of the encoder and decoder can potentially provide a better performance. However, in our real estate valuation experiment we find that linear encoder and decoder model is already good enough.

To illustrate this, we adopt one-hidden-layer feedforward neural network architecture for both encoder and decoder, which has input size $n$, hidden size $n$ and output size $n$, and the activation functions used by the hidden layer and output layer are ReLU and identity function, respectively. Then we set $\eta = 0.001$ and repeat the real estate valuation experiment (with other configurations being the same). We observe the performance of our task-aware approach is worse in Fig. 7 than in Fig. 5. The explanation is, due to the unnecessary complexity of the encoder and decoder, it is harder to learn good encoder and decoder parameters heuristically. Thus, we conclude that using a simple encoder and decoder, as also observed for the other datasets, is sufficient for high task accuracy.

### A.11 Experiment with High-dimensional Data

---

[3]The privacy-agnostic approach also achieves the best performance on average under the chosen $Z$'s.

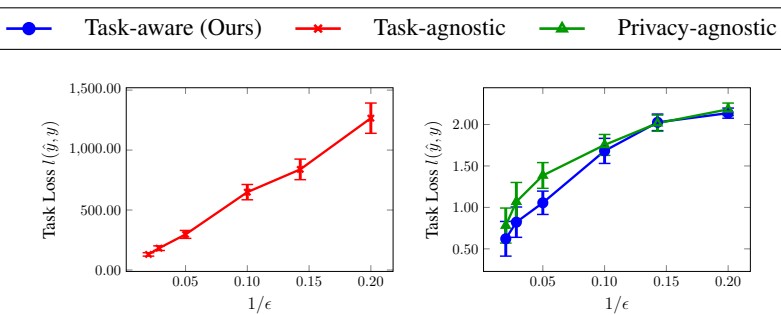

Figure 8: Task loss $l(\hat{y}, y)$ under different LDP budgets for handwritten digit recognition.

To illustrate our task-aware approach in Section 4.2 also works well for high-dimensional data, such as image data, we consider a handwritten digit recognition problem with well known MNIST dataset LeCun et al. (1998). Here, $x \in \mathbb{R}^{784}$ represents a $28 \times 28$ image of handwritten digit. And $y \in \{0, 1, \cdots, 9\}$ is a discrete variable represents the digit in the image. We first train a linear classification model using the ground truth $x$ and $y$, to serve as our task function $f$. Then we aim to minimize the cross-entropy loss of $\hat{y}$ and $y$, with linear encoder and decoder. We use $Z = 10$ for both our task-aware approach and the privacy-agnostic approach.

Fig. 5 shows the evaluation result. Since the task loss $\mathcal{L}$ of the task-agnostic approach is much larger than the other two approaches, we put it in a separate sub-figure on the left. On the right, we can see our task-aware approach always outperforms the privacy-agnostic approach on overall task loss $\mathcal{L}$ under different LDP budgets, which demonstrates the effectiveness of our proposed solution. The maximum improvement against the privacy-agnostic approach is $23.8\%$ ($\epsilon = 20$).

