# OpenReview forum: "Task-aware Privacy Preservation for Multi-dimensional Data"
_ICLR.cc/2022/Conference — ICLR 2022 Submitted_

### Official Review · Reviewer_mTuz · 2021-11-01

**Correctness:** 4
**Technical Novelty And Significance:** 3
**Empirical Novelty And Significance:** 3
**Recommendation:** 6
**Confidence:** 3

**Main Review:**

I think this paper addresses an important problem which is improving the privacy-utility trade-off in LDP where there are multiple users' data.
The proposed approaches solves the problem in an efficient way. The motivation is clear, it is well-structured and well-written. In my opinion, there are three things that can improve this paper. i) The paper presents a heuristic learning algorithm, but the analysis of the task-aware privacy preservation problem for approximate LDP would be more interesting. ii) Comparison with the state of the art approaches both in text and experiments could be improved. In the current version, the difference between the previously proposed methods is not so clear. iii) The results on larger datasets and the result in terms of accuracy.



**Summary Of The Paper:**

This paper proposes a task aware local-DP approach to improve the performance on multi-dimensional data with same level of privacy. This approach is based on encoder-decoder framework that perturbs only the task relevant encoding instead of the raw user data that generally causes less noise addition and improves the task accuracy. Besides, the paper provide a heuristic learning algorithm for more general settings. The proposed approach is compared with task agnostic and privacy agnostic approaches on different datasets and the results show the proposed method outperformes them on overall task loss under different privacy budgets.

**Summary Of The Review:**

This paper considers and important problem and brings an efficient solution. It is well-written. The experiments shows the performance improvement of the proposed method. However, the experiements could be extended on larger datasets and they can be compared with SOTA.

---

> ### Author Response · Authors · 2021-11-13
> **Reply to Reviewer mTuz**
>
> We thank the reviewer for the insightful comments.
>
> 1. Analysis for approximate LDP will be our future work, as we have mentioned in our "Conclusion and Future Work" section.
>
> 2. Please see our overall comment for comparisons with papers provided by other reviewers.
>
> 3.  High-dimensional experiment is added in the appendix A.11.

---

### Official Review · Reviewer_S4hg · 2021-11-02

**Correctness:** 3
**Technical Novelty And Significance:** 2
**Empirical Novelty And Significance:** Not applicable
**Recommendation:** 5
**Confidence:** 3

**Main Review:**

Strengths:
1. The problem of improving utility and privacy trade-off in local differentially private ML is an important practical problem.
2. For linear encoder and decoder, and MSE loss setting, an analytical near-optimal solution is provided. The comparisons among task-aware approach and the benchmark approaches give an intuitive interpretation of why the parameters obtained from task-aware approach are better.


My main concern in this paper is whether the proposed method can preserve local DP.
1. The local DP setting is not clearly given, which I mean, interactive [1] or non-interactive setting [2]?
2. Based on my understanding of this paper,  Algorithm 1 needs $N_{epochs}$ to find the encoder, decoder and sensitivity. During this interactive proposes, the algorithm continues to query the sensitive user data samples. I think the composition theorem should be considered.
3. For achieving private local data representation,  Wang et al. in [3] also proposed a lightweight privacy-preserving mechanism consisting of arbitrary data nullification and DP noise addition. Compared with [3], I find Algorithm 1 follows a quite similar process when learning the model parameters. Since [3] introduces data nullification and adversary training, their method may be more efficient and better than this paper in terms of communication and utility. Please provide a comparison with [3].


[1] Joseph, Matthew, Jieming Mao, Seth Neel, and Aaron Roth. "The role of interactivity in local differential privacy." In 2019 IEEE 60th Annual Symposium on Foundations of Computer Science (FOCS), pp. 94-105. IEEE, 2019.
[2] Wang, D., Gaboardi, M., Smith, A., & Xu, J. (2020). Empirical Risk Minimization in the Non-interactive Local Model of Differential Privacy. Journal of machine learning research, 21(200).
[3] Wang, Ji, Jianguo Zhang, Weidong Bao, Xiaomin Zhu, Bokai Cao, and Philip S. Yu. "Not just privacy: Improving performance of private deep learning in mobile cloud." In Proceedings of the 24th ACM SIGKDD International Conference on Knowledge Discovery & Data Mining, pp. 2407-2416. 2018.

**Summary Of The Paper:**

This paper proposes a task-aware local DP method to improve the privacy andutility trade-off for multi-dimensional user data. Also, a analytical near-optimal solution for a general linear encoder-decoder model and MSE loss is provided. For neural network cases (i.e., nonlinear encoder-decoder), the authors present a heuristic learning algorithm to get the model parameters. The experiments results demonstrate the effectiveness of the proposed methods.

**Summary Of The Review:**

This paper proposed a good analysis for task-aware local DP.  However, my main concern is the setting of local DP.

---

> ### Author Response · Authors · 2021-11-13
> **Reply to Reviewer S4hg**
>
> We thank the reviewer for the insightful comments.
>
> 1-2. Interactivity: Our model is non-interactive. Please see point 2 of our overall comment about offline/online. Since ground truth data is already exposed and used for the training at a centralized server, there's no need to query user data multiple times. $N_\text{epochs}$ is the number of training epochs needed at the centralized server to learn good representations.
>
> 3. We think [Wang18] has a key difference compared to our work (as our reply to the reviewer's point 1-2): Our cloud model possesses ground truth data, and hence after training it will find a better representation. [Wang18] doesn't have such a setting and hence the corresponding Algorithm 1 is centered around providing noisy data to the server.

---

### Official Review · Reviewer_mz4E · 2021-11-02

**Correctness:** 3
**Technical Novelty And Significance:** 2
**Empirical Novelty And Significance:** 2
**Recommendation:** 5
**Confidence:** 4

**Main Review:**

The curse of dimensionality and the problems it creates for utility of DP mechanisms is an important problem that has been studied for a long time. The approach this paper take's creates the following concerns:

1. The main idea of this  paper  which is dimensionality reduction, has already been proposed before, in numerous contexts such as random projections [1] or sparse regression [2]. The idea of mixing this with task-awareness and considering how a model would use features and adding noise only to features that are used has also been studied before [3]. It would be great if the paper quantifies its differences with these prior work/missing citations better.

2. How much overhead does this add to the user's side, in terms of latency and computation complexity? It will help to know how much extra latency gets added compared to doing normal local DP. The user seems to need to be able to carry out the encoder operation.

3. How much of this method's success depend on a correct analysis of what feature is related to the  task? How easy or hard is it to get such details? Some more clarity in explaining this would certainly help readability. It seems like the effectiveness of this method relates to how well related features can be extracted, and what this model is learning and if it really relates to how this data would be used in the future for other training tasks. This could even circle back to interpretability and how different models might use different features.

4. Although the paper claims effectiveness in high-dimensonal cases, the experiments  create the question "how well would this method scale in the presence of real-world, high dimensional data?", like medical images which are often large in size. The datasets used here are very small compared to real-life scenarios.

[1] Kenthapadi K, Korolova A, Mironov I, Mishra N. Privacy via the Johnson-Lindenstrauss Transform. Journal of Privacy and Confidentiality. 2013;5(1):39-71.

[2] Thakurta AG, Smith A. Differentially private feature selection via stability arguments, and the robustness of the lasso. InConference on Learning Theory 2013 Jun 13 (pp. 819-850). PMLR.

[3] Mireshghallah F, Taram M, Jalali A, Elthakeb AT, Tullsen D, Esmaeilzadeh H. A principled approach to learning stochastic representations for privacy in deep neural inference. arXiv preprint arXiv:2003.12154. 2020 Mar.

**Summary Of The Paper:**

The paper addresses the problem of large loss in utility due to the addition of noise in a local-DP setup, especially when working with high dimensional data. To mitigate this problem, the paper proposes an encoder-decoder setup in which noise is applied through the encoder, in a lower-dimension to decrease its effect on utility. Then  Laplace random noise is applied to each point's encoding in the latent space. The training  of the autoencoder is done over an offline phase on public data and with respect to the main task that the data is to be used for.  The approach is then evaluated empirically.

**Summary Of The Review:**


The main reasons for my recommendation are:

1. similarity to prior work, the novelty of this work and it's benefits are not well identified

2. How useful the extracted dimensions would be in high dimensions and for models that might use different set of features is not clear.

---

> ### Author Response · Authors · 2021-11-13
> **Reply to Reviewer mz4E**
>
> We thank the reviewer for the insightful comments.
>
> 1.
> i) About random projection, like in [Kenthapadi13]: Since for random projection there is randomness in selecting the projection (besides noise perturbation for privacy), the performance mainly relies on the probability of the selected projection being good (i.e., more task-related). However, the key benefit of our task-aware approach is we **directly optimize** for a good projection either analytically or numerically.
> ii) [Thakurta13] is centered around stability, and is different from the problem of finding a task-aware presentation under a given privacy budget.
> iii) Our problem is different from [Mireshghallah20] because of point 2,3,4 in our overall comment.
> 2. Indeed, the complexity of the transformation for the encoder will be a factor to consider at the user's end. And there's additional latency which depends on how fast the user's device is. To make the computation overhead small we limit the encoder to linear and 1-hidden layer neural networks. What's more, the frequency of data collection is in general low (for example, in experiment 1 we only need to collect the user's power consumption daily). So we believe the improvement of task-accuracy for task-aware representation overweighs the impact of latency.
> 3. We think data interpretability is orthogonal to this work. Both our task-aware approach and task-agnostic benchmark approach will select the top-Z features, which we don't expect to be very different. However, the major difference is our task-aware approach will scale the features differently based on their importance and then add noise, while the task-agnostic benchmark approach will keep the scales randomly (for example, uniformly) and hence is not able to minimize task loss (though it is still better than perturbing all the features).
> 4. High-dimensional experiment is added in Appendix A.11.

---

### Official Review · Reviewer_r5c5 · 2021-11-04

**Correctness:** 2
**Technical Novelty And Significance:** 2
**Empirical Novelty And Significance:** 2
**Recommendation:** 3
**Confidence:** 5

**Main Review:**

 Strengths
+ The problem formulation is simple and easy to follow to get the idea of the proposed method.
+ They provide an intensive and rigorous analysis for an analytical near-optimal solution for a linear setting and MSE task loss in terms of task-aware privacy preservation analysis.

Weaknesses
-	First, the general idea of the paper is interesting, but it seems to be incremental and at an early stage of this work. The novelty is limited. The idea of redistributing the noise across input features has been extensively studied, for instance, with autoencoder-decoder and layer-wise relevant propagation, and forward derivatives (Phan et al., 2017; 2019). These references should be considered as baselines for comparison to highlight the novelty of the proposed approach. However, these critical references are missing.
-	Second, the reviewer does not understand why we need LDP here and why not centralized DP? What is the threat model? Training the autoencoder-decoder would need a centralized server to gather all the data tuples together. If so, there is no need for LDP.
-	Third, how practical is Assumption 1? It is not clear at all in the current writing. Also, it is unclear to the reviewer that why Propositions 1-4 result in an optimal encoder and decoder design that preserves $\epsilon$-LDP.
-	In the analysis, a linear model with an MSE task loss is used in the analysis and experiments, which may not be easy to extend to other complex tasks, such as deep learning.
-	Experimental results can be improved. The datasets used in the experiments still have a limited number of dimensions, such as dim=24 in power consumption, dim=6 in real estate valuation, and dim=30 in breast cancer detection. It would be interesting if the work can be done with the more complex datasets and learning tasks, such as image datasets with thousands of dimensions and deep learning tasks for example. Adaptive mechanisms such as (Phan et al., 2017; 2019; 2020) and other related works listed below can be considered as baseline approaches for comparison.
-	The compared benchmark approaches are unclear. There is no place to define or refer to the compared method. The reviewer needs to guess what it means. Also, there is a limited number of comparisons, which makes the paper unconvincing.

NhatHai Phan, Xintao Wu, Han Hu, Dejing Dou. Adaptive Laplace Mechanism: Differential Privacy Preservation in Deep Learning. IEEE ICDM'17.

NhatHai Phan, Minh Vu, Yang Liu, Ruoming Jin, Xintao Wu, Dejing Dou, and My T. Thai. Heterogeneous Gaussian Mechanism: Preserving Differential Privacy in Deep Learning with Provable Robustness. IJCAI'19.

**Summary Of The Paper:**

This paper provides task-aware privacy preservation to improve the task performance for multi-dimensional user data for deploying a trained model. The paper’s spirit is to use an encoder-decoder framework with a Laplace noise added to learn a task-relevant but LDP privacy preserved latent representation of user data. The author also provides an analytical near-optimal solution for a linear setting with mean-squared error task loss. They show the effectiveness of their method via experiments on three real-world datasets.

**Summary Of The Review:**

The general idea of the paper is interesting, but it seems to be an early stage of this work. Investigating the work with more complex datasets and tasks and more baseline comparisons would make the paper more convincing. There are a lot baseline of LDP or task-aware work to compare with the work, for example
[1] Wang, N., Xiao, X., Yang, Y., Zhao, J., Hui, S. C., Shin, H., ... & Yu, G. (2019, April). Collecting and analyzing multidimensional data with local differential privacy. In 2019 IEEE 35th International Conference on Data Engineering (ICDE) (pp. 638-649). IEEE.
[2] Liu, R., Cao, Y., Yoshikawa, M., & Chen, H. (2020, September). Fedsel: Federated sgd under local differential privacy with top-k dimension selection. In International Conference on Database Systems for Advanced Applications (pp. 485-501). Springer, Cham.

---

> ### Author Response · Authors · 2021-11-13
> **Reply to Reviewer r5c5**
>
> We thank the reviewer for the insightful comments.
>
> 1. We were aware of the series of work provided by the reviewer and **already cited** [Phan16] in our original manuscript (point 3 of "Related Work" on Page 2). Similar to [Phan16], [Phan17] and [Phan19] are relevant but not closely related to our work, because of points 2,3,4 for [Phan17] and points 1,2,3,4 for [Phan19] of our overall comment. We have added [Phan17] and [Phan19] to our citation list.
> 2. LDP vs centralized DP: it is mainly because of our problem setting, as mentioned in point 2 of our overall comment.
> 3. Assumption 1 is a strong assumption, as we pointed out right after. 1) If it indeed holds (which could be rare), proposition 1-4 will determine the optimal encoder and decoder, the proofs of which are given in the appendix; 2) If it doesn't hold (which is more likely), our Theorem 1 is able to **bound the optimal task loss in a region**, which also tells how much we will lose at most by assuming Assumption 1 is true. The difference will further depend on how much Assumption 1 is violated (i.e., the value of $r_\text{max} - r_\text{min}$).
> 4. We believe linear analysis is applicable in many real-world scenarios because of point 4 of our overall comment. It doesn't apply to DNNs but our Algorithm 1 will be the approach. Furthermore, our experiment 3 and an additional experiment in A.10 also adopt a 1-hidden layer neural network which is nonlinear but still compute-efficient.
> 5. High-dimensional experiment is added in the Appendix A.11.
> i) [Wang2019a] was already cited in our "Related Work" section, point 1. We listed it as one of the three similar works to our study, and discussed why our approach is different.
> ii) [Liu20] does not apply since it is about adding noise to the gradient, see point 4 of our overall comment.
> 6. The definitions of the benchmarks are in the "Benchmarks" paragraph on Page 4.

---

### Author Response · Authors · 2021-11-13
**Overall Comment**

We sincerely thank the reviewers for all their valuable comments.

First, we have added a **high-dimensional experiment** in Appendix A.11, which confirms our results. Second, we now clarify some common misunderstandings about our problem setting, which is quite different from the papers provided by the reviewers.

As an example, consider a scenario where raw user data $x$ must be sent to a central server for inference with a pre-trained ML task model $y = f(x)$. We first train a privacy-preserving encoder/decoder **offline** using data from a set of consenting volunteers. Then, we deploy the trained/encoder decoder **online** on new user data while preserving LDP. The key technical novelty is our matrix analysis that calculates the sensitivity of the task function $f$ to noise perturbations to **greatly improve the privacy-utility trade-off** compared to benchmarks. As such, this should clarify the following key features of our setting.

1) We don't add noise to the **gradient** to preserve DP/LDP during the training of deep neural networks. Instead, our approach in section 4.2 finds a good data **representation** and adds noise to it to preserve LDP.
2) To determine such a representation, we train the model in a centralized and offline manner, where some ground truth user data is needed, as we explained in the last few lines in "Related Work" on page 2. But in the online case the centralized server will **never know** ground truth data since we have already deployed our LDP-preserving encoder/decoder. In summary, we don't guarantee privacy offline but guarantee privacy online, and hence we don't need to collect perturbed user data interactively for model training. Such a setting is practical since a few consenting volunteers’ data can be used for offline model training.
3) Since there's ground truth data during our training phase, we can train our task model (a neural network) first without privacy consideration, and then train the encoder and decoder with a fixed task model to preserve LDP. This allows us to determine the sensitivity of the ultimate task loss to noise perturbations, which only appears in the second part.
4) We study LDP rather than DP. Though DP and LDP share some similarities, a realistic difference is that part of the LDP model, i.e., encoder, is deployed at the user's end and may need to be lightweight. This explanation has been included in our manuscript.

Furthermore, we have made the following changes to our manuscript:
1) We have added the papers provided by the reviewers in our "Related Work" section as well as in our citation list. A detailed comparison or explanation for each paper is provided in our replies, and we are happy to update this section with the reviewer’s suggestion.
2) We briefly discussed why the encoder should be lightweight (e.g. compute-efficient) at the end of paragraph 2 of section 3.  Imagine it is run on a user’s mobile phone, for example.
3) We added a new experiment with high-dimensional data (MNIST) in appendix A.11, which demonstrates our task-aware approach works for high-dimensional data as well.

We now reply to each reviewer separately, and are happy to address the reviewer's further questions/concerns in the discussion phase.

---

### Decision · Program_Chairs · 2022-01-20

**Decision:**

Reject

**Comment:**

This work considers the problem of how to predict on sensitive user points while preserving their privacy. It proposes a fairly straightforward way to create a local randomizer that optimizes loss for a given model subject to preserving LDP. The work also gives theoretical analysis of the randomizer for least squares linear regression.
The problem formulation is different from the standard LDP framework where privacy of training data points needs to be preserved. The submission does not motivate this setting and I don't see a good motivation for this problem either. More importantly, it does not sufficiently emphasize that the problem is entirely different from prior work. Indeed all reviewers were confused about various aspects of comparison with previous work. Therefore, in my opinion, the submission is not sufficiently well motivated and clearly presented to be accepted.